# An Improved C_0_ FE Model for the Sandwich Lattice Composite Panel

**DOI:** 10.3390/polym13234200

**Published:** 2021-11-30

**Authors:** Junqing Hong, Chunyan Shen, Weiqing Liu, Hai Fang, Laiyun Yang

**Affiliations:** 1School of Transport and Civil Engineering, Nantong University, Nantong 226019, China; 2College of Civil Engineering, Nanjing Tech University, Nanjing 211816, China; scy_123123@163.com (C.S.); wqliu@njtech.edu.cn (W.L.); 201761101657@njtech.edu.cn (L.Y.)

**Keywords:** sandwich panel, lattice composite panel, zigzag theory, finite element, stress continuity

## Abstract

Combining the improved C_0_ plate element using high-order zigzag theories and the beam element degenerated from the plate element, a type of analysis model for the sandwich lattice composite panel was developed. Compared with the actual test results including the mid-span deflections and the surface sheet normal stresses, the outstanding of that method was presented through numeric calculation. The results showed that the model has great potential to become an excellent and highly efficient analysis and design tool for sandwich lattice composite panel to avoid the conventional three-dimension hybrid element model, which usually may lead to the complex program establishment, and the coupling degrees of freedom among the different types of elements.

## 1. Introduction

Composite materials and structures based on fiber-reinforced polymer (FRP) are widely used in many areas, such as aerospace, transportation, and building due to their excellent performance qualities of high strength, lightweight, anti-corrosion, easy design, and multi-forms.

One of the abovementioned is the laminated sandwich plate (LSP), which is composed of lamina face sheets stacked with each other and thick core materials, shows superior mechanical properties, and is applied in engineering practices [1,2,3,4]. Face sheets are mainly made of fiber materials such as glass fiber-reinforced polymer (GFRP), carbon fiber-reinforced polymer (CFRP), and core materials that include polyvinyl chloride (PVC) foam, polyurethane (PU) foam, balsa wood, and bamboo. Plenty of investigations have been conducted on the mechanical properties of LSP such as flexure, pressure, fatigue, delamination, etc. [5,6,7,8,9,10,11]. The research showed that the delamination usually occurred at the bonding interface between face sheets and core materials due to poor constraint of the foam core. Weiqing Liu, Hai Fang, et al. [12,13,14,15] presented a kind of sandwich-latticed composite panel (SLCP) manufactured by vacuum-infusing resin process, as shown in Figure 1. The lattice webs were added into the foam core to improve the peel resistance between face sheet and core surface. However, the crisscross reinforced webs destroyed the continuity of core and led to a complicated analysis model which cannot be modeled as LSP simply. Therefore, this paper presented a type of analysis model to calculate SLCP accurately and efficiently.

LSP can be calculated by several methods, including: (i) Equivalent modulus analysis [16] based on the classical laminate theory, which is hard for the complicated SLCP due to existing webs, which neglect many details of cores and webs and omit the transverse shear stress. (ii) Three-dimension finite element method (FEM), which is usually considered to be omnipotent for complex model construction. However, several issues may block its convenient application, including huge programming workload, calculation efficiency, degree-of-freedom (DOF) coupling among the different types of elements, and inflexible element displacement mode even if general commercial finite-element software is used. (iii) Plate theory is another potential choice for the SLCP analysis, which simplifies the LSP calculation. Various models have been proposed rested on the single-layer theory, layer-wise theory, and zigzag theory respectively.

Kirchhoff’s classic laminate plate theory (CLT) [17] was the first comprehensive theory developed for plates but it did not take the transverse shear strains into consideration. The single-layer theory is also known as first-order shear deformation theory (FSDT) and the disadvantage is lack of a shear correction factor to predict the actual parabolic variation of shear stress and shear locking phenomenon for thin plates, which is still being investigated now [18,19]. To overcome the shortcomings of FSDT, higher order shear deformation theory (HSDT) [20,21] was proposed by Reddy, which is accurate and accounts for the transverse shear deformation and the transverse shear traction-free conditions on the top and the bottom surfaces of the plate without shear correction factor. However, the C_1_ continuity issue along the interelement boundary promotes the theory development to some degree. The layer-wise theories were presented to surmount that issue. Discrete layer theories were concerned and proposed by A.Toledano, H.Murakami [22], and Reddy [23], which take unknown displacement components through all the layer interfaces. This plate theory possesses good performance, but the numbers of unknowns increase rapidly with the increase of layers that, in fact, lead to huge unaccepted calculations. So, zigzag theories (ZZT, known as refined plate theories) were developed by H. Murakami [24], S.P. Lee, et al. [25] for solving the aforementioned question though the unknowns at different interfaces linking to those at the reference plan unknowns. Improved versions about these theories have been suggested continuously. However, the zigzag theory still faced the C_1_ continuity issue of the transverse displacement at the nodes by FEM. Combining the benefits of the discrete layer wise and higher order zigzag theories (HOZT), sub-laminate models and penalty stiffness multiplier were propounded to overcome C_1_ continuity successively for LSP [26,27,28,29,30].

In fact, the improved C_0_ FE model could receive accurate results efficiently [31,32,33,34]. Chalak et al. [35,36,37,38,39] proposed an improved C_0_ finite element model for LSP analysis with a soft, compressible core using HOZT. In this model, the in-plane displacement fields as a combination of a linear zigzag function with different slopes at each layer and cubically varying function over the entire thickness are assumed. The out-of-transverse displacement within the core and the surface of face sheets is presumed to be quadratic and constant. This model satisfies the transverse shear stress continuity conditions at the layer interfaces and the conditions of zero transverse shear stress at the top and bottom of the plate, and has made significant contributions: (i) overcoming the C_1_ continuity problem associated with HOZT to implement a C_0_ formulation, (ii) possessing great effects on the core compressibility in the formulation, and (iii) eliminating the requirement of using a penalty multiplier in the stiffness matrix formation. The model can also be applied for the ordinary and medium-thick LSP.

In this paper, the improved C_0_ plate finite element with soft compressible core made use of high-order zigzag theories and the beam element, which is the degeneration form of the above plate element (IC_0_FEM-HOZT). This finite element was combined to calculate the sandwich lattice composite panel under the out-of-transverse quasi-static loading. Therefore, a type of high efficient analysis method was developed rather than the ordinary three-dimension FEM composed of conventional shell or plate, beam, and solid elements. Nine-nodes-isoparametric-quadratic plate element was used for the face sheets and core. Three-nodes-isoparametric-quadratic beam element, regarded as a type of plate model degeneration, was adopted for the lattice webs cutting apart the core. The spatial occupied core areas by the web volumes can be subtracted by the same virtualized volume web with the core material, or even omitted since the web is very thin and the elastic modulus is far higher than that of the core material. The outstanding performance of that method was presented through numeric calculation compared with the actual test results, including the mid-span deflections and the surface feet normal stresses.

## 2. Mathematic Model

### 2.1. Model Displacements

The in-plane displacement fields [28] for the plate, shown in Figure 2, were chosen as follows:(1)$$U=u_{0}+z\theta_{x}+\sum_{i=1}^{n_{u}-1}\left(z-z_{i}^{u}\right)H\left(z-z_{i}^{u}\right)\alpha_{xu}^{i}+\sum_{j=1}^{n_{l}-1}\left(z-z_{j}^{l}\right)H\left(-z+z_{j}^{l}\right)\alpha_{xl}^{j}+\beta_{x}z^{2}+\eta_{x}z^{3}$$
(2)$$V=v_{0}+z\theta_{y}+\sum_{i=1}^{n_{u}-1}\left(z-z_{i}^{u}\right)H\left(z-z_{i}^{u}\right)\alpha_{yu}^{i}+\sum_{j=1}^{n_{l}-1}\left(z-z_{j}^{l}\right)H\left(-z+z_{j}^{l}\right)\alpha_{yl}^{j}+\beta_{y}z^{2}+\eta_{y}z^{3}$$
where *u*_0_ and *v*_0_ are on behalf of the in-plane displacements of any point at the midsurface, *θ_x_* and *θ_y_* are the rotations of normal to the middle plane about the *y*- and *x*-axis respectively; *n_u_* and *n_l_* are the number of upper and lower layers respectively; $\beta_{x}$, $\beta_{y}$, $\eta_{x}$, $\eta_{y}$ are the higher unknowns; $\alpha_{xu}^{i}$, $\alpha_{yu}^{i}$, $\alpha_{xl}^{j}$, $\alpha_{yl}^{j}$ are the slopes of i th and j th layer corresponding to upper and lower layers respectively, and $H\left(z-z_{i}^{u}\right)$ and $H\left(-z+z_{j}^{l}\right)$ are the unit step functions.

The out-of-transverse displacement was assumed to vary quadratically through the core thickness and be constant over the face sheets, which is expressed as Equation (3) and shown in Figure 3:(3)$$W=l_{1}w_{u}+l_{2}w_{0}+l_{3}w_{l},or=w_{u},or=w_{0}$$
where $w_{u}$, $w_{0}$, and $w_{l}$ are the values of the out-of-transverse displacement at the top layer, middle layer, and bottom layer of the core respectively, and *l*_1_, *l*_2_, *l*_3_, are Lagrangian interpolation functions in the thickness co-ordinate as defined in [35,36].

### 2.2. Model of Constitutive Relations

The constitutive relationship of any *k*th orthotropic layer for plate or beam having any fiber orientation with the respect to structural axes system (*x*-*y*-*z* or *x*-*z*) is depicted as
(4)$$\left\{\sigma \right\}={\left[Q\right]}_{k}\left\{\bar{\varepsilon }\right\}$$
where {σ} and $\left\{\bar{\varepsilon }\right\}$ are the stress vector and the strain vector. ${\left[Q\right]}_{k}$ is the transformed rigidity matrix of *k*th layer that can be evaluated with the material properties (*E*, elastic modulus; *v,* poisson ratio; *G*, shear modulus) and fiber orientation of *k*th layer on mechanics of composite structure.

The detailed Equation (4) for plate is:$$\left\{\begin{array}{c} \sigma_{x}\\ \sigma_{y}\\ \sigma_{z}\\ \tau_{xy}\\ \tau_{xz}\\ \tau_{yz} \end{array}\right\}={\left[\begin{array}{cccccc} Q_{11} & Q_{12} & Q_{13} & Q_{14} & 0 & 0\\ Q_{21} & Q_{22} & Q_{23} & Q_{24} & 0 & 0\\ Q_{31} & Q_{32} & Q_{33} & Q_{34} & 0 & 0\\ Q_{41} & Q_{42} & Q_{43} & Q_{44} & 0 & 0\\ 0 & 0 & 0 & 0 & Q_{55} & Q_{56}\\ 0 & 0 & 0 & 0 & Q_{65} & Q_{66} \end{array}\right]}_{k}\left\{\begin{array}{c} {\bar{\varepsilon }}_{x}\\ {\bar{\varepsilon }}_{y}\\ {\bar{\varepsilon }}_{z}\\ {\bar{\gamma }}_{xy}\\ {\bar{\gamma }}_{xz}\\ {\bar{\gamma }}_{yz} \end{array}\right\}$$

The detailed Equation (4) for beam is:$$\left\{\begin{array}{c} \sigma_{x}\\ \sigma_{z}\\ \tau_{xz} \end{array}\right\}={\left[\begin{array}{ccc} Q_{11} & Q_{13} & 0\\ Q_{31} & Q_{33} & 0\\ 0 & 0 & Q_{55} \end{array}\right]}_{k}\left\{\begin{array}{c} {\bar{\varepsilon }}_{x}\\ {\bar{\varepsilon }}_{z}\\ {\bar{\gamma }}_{xz} \end{array}\right\}$$

On the conditions of zero transverse shear stress at the top and bottom surfaces of the plate or beam, and the transverse shear stress continuity at the interfaces between the layers with the condition, $u=u_{u}$ and $v=v_{u}$ at the top, and $u=u_{l}$ and $v=v_{l}$ at the bottom of the plate, $\beta_{x}$, $\eta_{x}$, $\beta_{y}$, $\eta_{y}$, $\alpha_{xu}^{i}$, $\alpha_{xl}^{i}$, $\alpha_{yu}^{i}$, $\alpha_{yl}^{i}$, $\partial w_{u}/\partial x$, $\partial w_{l}/\partial x$, $\partial w_{u}/\partial y$, and $\partial w_{l}/\partial y$ can be expressed by the displacements $u_{0}$, $v_{0}$, $\theta_{x}$, $\theta_{y}$, $u_{u}$, $u_{l}$, $v_{u}$, and $v_{l}$ as
(5){B}=[A]{α}
where for plate as
$$\begin{array}{c} {\left\{B\right\}}_{plate}={\left\{\begin{array}{c} \beta_{x} \end{array}\begin{array}{c} \eta_{x}\begin{array}{c} \beta_{y} \end{array}\begin{array}{c} \eta_{y}\alpha_{xu}^{1}\alpha_{xu}^{2}\cdots \alpha_{xu}^{n_{u}-1}\alpha_{xl}^{1}\alpha_{xl}^{2}\cdots \alpha_{xl}^{n_{l}-1}\alpha_{yu}^{1}\alpha_{yu}^{2}\cdots \alpha_{yu}^{n_{u}-1}\alpha_{yl}^{1}\alpha_{yl}^{2}\cdots \alpha_{yl}^{n_{l}-1}\left(\partial w_{u}/\partial x\right)\left(\partial w_{u}/\partial y\right)\left(\partial w_{l}/\partial x\right)\left(\partial w_{l}/\partial y\right) \end{array} \end{array}\right\}}^{T}\\ {\left\{\alpha \right\}}_{plate}={\left\{\begin{array}{c} u_{0}v_{0}\theta_{x}\theta_{y}u_{u}v_{u}u_{l}v_{l} \end{array}\right\}}^{T} \end{array}$$

For beam as
$$\begin{array}{c} {\left\{B\right\}}_{beam}={\left\{\begin{array}{c} \beta_{x} \end{array}\begin{array}{c} \eta_{x}\begin{array}{c} \alpha_{xu}^{1}\alpha_{xu}^{2}\cdots \alpha_{xu}^{n_{u}-1}\alpha_{xl}^{1}\alpha_{xl}^{2}\cdots \alpha_{xl}^{n_{l}-1}\left(\partial w_{u}/\partial x\right)\left(\partial w_{l}/\partial x\right) \end{array} \end{array}\right\}}^{T},\\ {\left\{\alpha \right\}}_{beam}={\left\{\begin{array}{c} u_{0}\theta_{x}u_{u}u_{l} \end{array}\right\}}^{T} \end{array}$$

And elements of [A] are dependent on material properties. The last derivatives entries of the vector {B} help overcome the problem of C_1_ continuity as mentioned before, including last four items of the plate and last two items of the beam.

According to the above equations, the in-plane displacement fields for plate as given in Equations (1) and (2) may be expressed as

For plate
(6)$$U=b_{1}u_{0}+b_{2}v_{0}+b_{3}\theta_{x}+b_{4}\theta_{y}+b_{5}u_{u}+b_{6}v_{u}+b_{7}u_{l}+b_{8}v_{l}$$
(7)$$V=c_{1}u_{0}+c_{2}v_{0}+c_{3}\theta_{x}+c_{4}\theta_{y}+c_{5}u_{u}+c_{6}v_{u}+c_{7}u_{l}+c_{8}v_{l}$$

For beam
(8)$$U=b_{1}u_{0}+b_{3}\theta_{x}+b_{5}u_{u}+b_{7}u_{l}$$
where the coefficients *b_i_* and *c_i_* are functions of thickness coordinates, unit step functions, and material properties as defined in [35,36]. Equations (6)–(8) do not contain any first-order derivative terms of out-of-transverse displacements and avoid the requirements of C_1_ continuity efficiently without new field variables [28] and penalty method [40,41].

The generalized displacement vector {δ} for the plate and beam model can be presented as

For plate ${\left\{\delta \right\}}_{plate}=\left\{u_{0}v_{0}w_{0}\theta_{x}\theta_{y}u_{u}v_{u}w_{u}u_{l}v_{l}w_{l}\right\}$

For beam ${\left\{\delta \right\}}_{beam}=\left\{u_{0}w_{0}\theta_{x}u_{u}w_{u}u_{l}w_{l}\right\}$

With the linear constitutive relation and Equations (1)–(5), the strain field can be expressed by unknowns from the structural deformations as

For plate
(9)$${\left\{\bar{\delta }\right\}}_{plate}={\left\{\frac{\partial U}{\partial x}\frac{\partial V}{\partial y}\frac{\partial W}{\partial z}\left(\frac{\partial U}{\partial x}+\frac{\partial V}{\partial y}\right)\left(\frac{\partial U}{\partial z}+\frac{\partial W}{\partial x}\right)\left(\frac{\partial V}{\partial z}+\frac{\partial W}{\partial x}\right)\right\}}^{T}$$
where Equation (9) can be simplified as ${\left\{\bar{\delta }\right\}}_{plate}={\left[H\right]}_{plate}\left\{\delta \right\}$,

For beam
(10)$${\left\{\bar{\delta }\right\}}_{beam}={\left\{\frac{\partial U}{\partial x}\frac{\partial W}{\partial z}\left(\frac{\partial U}{\partial z}+\frac{\partial W}{\partial x}\right)\right\}}^{T}$$
where Equation (10) can be simplified as ${\left\{\bar{\delta }\right\}}_{beam}={\left[H\right]}_{beam}\left\{\delta \right\}$,

Where for plate
$$\begin{array}{c} {\left\{\varepsilon \right\}}_{plate}=\{u_{0}v_{0}w_{0}\theta_{x}\theta_{y}u_{u}v_{u}w_{u}u_{l}v_{l}w_{l}\left(\partial u_{0}/\partial x\right)\left(\partial u_{0}/\partial y\right)\left(\partial v_{0}/\partial x\right)\left(\partial v_{0}/\partial y\right)\left(\partial w_{0}/\partial x\right)\left(\partial w_{0}/\partial y\right)\\ \left(\partial \theta_{x}/\partial x\right)\left(\partial \theta_{x}/\partial y\right)\left(\partial \theta_{y}/\partial x\right)\left(\partial \theta_{y}/\partial y\right)\left(\partial u_{u}/\partial x\right)\left(\partial u_{u}/\partial y\right)\left(\partial v_{u}/\partial x\right)\left(\partial v_{u}/\partial y\right)\\ {\left(\partial w_{u}/\partial x\right)\left(\partial w_{u}/\partial y\right)\left(\partial u_{l}/\partial x\right)\left(\partial u_{l}/\partial y\right)\left(\partial v_{l}/\partial x\right)\left(\partial v_{l}/\partial y\right)\left(\partial w_{l}/\partial x\right)\left(\partial w_{l}/\partial y\right)\}}^{T} \end{array}$$

For beam
$${\left\{\varepsilon \right\}}_{beam}={\left\{u_{0}w_{0}\theta_{x}u_{u}w_{u}u_{l}w_{l}\left(\partial u_{0}/\partial x\right)\left(\partial w_{0}/\partial x\right)\left(\partial \theta_{x}/\partial x\right)\left(\partial u_{u}/\partial x\right)\left(\partial w_{u}/\partial x\right)\left(\partial u_{l}/\partial x\right)\left(\partial w_{l}/\partial x\right)\right\}}^{T}$$

And the elements of ${\left[H\right]}_{plate}$ and ${\left[H\right]}_{beam}$ are functions of z and unit-step functions, as given in [35,36]

The potential energy of the system can be expressed as
(11)$$\Pi_{e}=U_{s}-W_{est}$$
where *U_s_* is the strain energy and *W_est_* is the work due to the elemental out-of-transverse static load.

Equations (4), (9), and (10), *U_s_* can be presented by
(12)$$U_{s}=\frac{1}{2}\sum_{k=1}^{n}∭{\left\{\bar{\varepsilon }\right\}}^{T}{\left[Q\right]}_{k}\left\{\bar{\varepsilon }\right\}dxdydz=\frac{1}{2}\iint {\left\{\varepsilon \right\}}^{T}{\left[D\right]}_{plate}\left\{\varepsilon \right\}dxdy+\frac{1}{2}\int {\left\{\varepsilon \right\}}^{T}{\left[D\right]}_{beam}\left\{\varepsilon \right\}dx$$
where for plate
(13)$${\left[D\right]}_{plate}=\sum_{k=1}^{n}{\int \left[H\right]}_{plate}^{T}{\left[Q\right]}_{plate,k}{\left[H\right]}_{plate}dz$$

For beam
(14)$${\left[D\right]}_{beam}=\sum_{k=1}^{n}{\int \left[H\right]}_{beam}^{T}{\left[Q\right]}_{beam,k}{\left[H\right]}_{beam}dz$$

The work due to the elemental transverse static load *P* can be calculated by
(15)$$W_{est}=\iint Pwdxdy$$

To solve this problem, a nine-node quadratic element with 11 field variables $\left(u_{0},v_{0},w_{0},\theta_{x},\theta_{y},u_{u},v_{u},w_{u},u_{l},v_{l},w_{l}\right)$ per node was employed for the plate. A three-node quadratic element with seven field variables $\left(u_{0},w_{0},\theta_{x},u_{u},w_{u},u_{l},w_{l}\right)$ per node was employed for the beam, which coordinates the plate element conveniently. The generalized displacement vector at any point for any plate or beam can be expressed as
(16)$${\left\{\delta \right\}}_{plate,or,beam}=\sum_{i=1}^{n}N_{i}{\left\{\delta \right\}}_{i}$$
where ${\left\{\delta \right\}}_{plate}={\left\{u_{0},v_{0},w_{0},\theta_{x},\theta_{y},u_{u},v_{u},w_{u},u_{l},v_{l},w_{l}\right\}}^{T}$ for plate, and ${\left\{\delta \right\}}_{beam}={\left\{u_{0},w_{0},\theta_{x},u_{u},w_{u},u_{l},w_{l}\right\}}^{T}$ for beam. ${\left\{\delta \right\}}_{i}$ is the displacement vector corresponding to node *i* of plate or beam element; $N_{i}$ is the shape function associated with node *i* and *n* is the number of nodes per element, that is, nine for the plate or three for the beam.

With the help of Equation (16), the strain vector {ε} for the plate or beam can be expressed in terms of the generalized displacement vector ${\left\{\delta \right\}}_{plate}$ or ${\left\{\delta \right\}}_{beam}$ as
(17)$${\left\{\varepsilon \right\}}_{plate}={\left[B\right]}_{plate}{\left\{\delta \right\}}_{plate}$$
(18)$${\left\{\varepsilon \right\}}_{beam}={\left[B\right]}_{beam}{\left\{\delta \right\}}_{beam}$$
where ${\left[B\right]}_{plate}$ or ${\left[B\right]}_{beam}$ are the strain-displacement matrices in the Cartesian coordinate system.

The elemental potential energy as given in Equation (11) can be rewritten as
(19)$$\begin{array}{ll} \prod_{e} & =\frac{1}{2}\iint {\left\{\delta \right\}}_{plate}^{T}{\left[B\right]}_{plate}^{T}{\left[D\right]}_{plate}{\left[B\right]}_{plate}{\left\{\delta \right\}}_{plate}dxdy+\frac{1}{2}\int {\left\{\delta \right\}}_{beam}^{T}{\left[B\right]}_{beam}^{T}{\left[D\right]}_{beam}{\left[B\right]}_{beam}{\left\{\delta \right\}}_{beam}dx-\frac{1}{2}{\iint \left\{\delta \right\}}_{plate}^{T}{\left[B\right]}^{T}{\left[N^{w}\right]}^{T}Pdxdy\\ & =\frac{1}{2}{\left\{\delta \right\}}_{plate}^{T}\left[K_{e}\right]{\left\{\delta \right\}}_{plate}-\frac{1}{2}{\left\{\delta \right\}}_{plate}^{T}\left\{P_{e}\right\} \end{array}$$
where
(20)$$\left[K_{e}\right]=\iint {\left[B\right]}_{plate}^{T}{\left[D\right]}_{plate}{\left[B\right]}_{plate}dxdy+\int {\left[B\right]}_{beam}^{T}{\left[D\right]}_{beam}{\left[B\right]}_{beam}dx$$
(21)$$\left\{P_{e}\right\}=\iint {\left[N^{w}\right]}^{T}Pdxdy$$
where ${\left[N^{w}\right]}^{T}$ is the shape function like matrix with non-zero terms associated only with the corresponding out-of-transverse nodal displacements. Though the beam element node lacks partial field variables, the existing ones are all included in the plate-element-node field variables and are easy to extend to the plate’s.

In accordance with Principle of Minimum Potential Energy, minimizing $\prod_{e}$ as given in Equation (19) with respect to ${\left\{\delta \right\}}_{plate}$, the equilibrium equation is
(22)$$\left[K_{e}\right]{\left\{\delta \right\}}_{plate}=\left\{P_{e}\right\}$$
where $\left[K_{e}\right]$ is the element stiffness matrix, and $\left\{P_{e}\right\}$ is the nodal load vector.

The global stiffness matrix was formed by taking the contribution of all the plate elements and beam elements. The formation of the global load vector for the whole SLCP was just formed in consideration of the plate elements that contain the all-beam nodes. Then, the global linear simultaneous equations were formed and solved for the SLCP incorporating appropriate boundary conditions. In order to improve the displacements-calculation efficiency by the FE model, the sparse-matrix technique was utilized to store the global stiffness matrix. The stresses were calculated with the constitutive relationship by using the condition of stress continuity as in Equation (5). Meanwhile, this model, combined with the improved C_0_ Zigzag plate model and its degeneration beam model, naturally circumvented the DOF coupling, made the beam and plate deformation compatible, and simplified the programming process.

## 3. Comparison Cases

To verify the effectiveness of the aforementioned improved C_0_ finite element method on the ground of HOZT (IC_0_FEM-HOZT), the corresponding test results were compared here. Two cases about SLCP flexural experiments in the literature are presented as the referenced examples in the following sections [14,15].

### 3.1. Case 1: Two-Way Simply Supported SLCP under the Concentrated Load

#### 3.1.1. Specimen and Experiment Introduction

Five two-way reinforced SLCPs, composed of GFRP face sheets, GFRP webs, and rigid polyurethane foam cores, by vacuum-assisted resin infusion process, were tested under the concentrated load, which was loaded by the hydraulic actuator. The face sheet was formed by 0°/90°GFRP clothes, and the web sheet consisted of −45°/45°GFRP clothes. That length and width of all SLCP specimens were 1000 mm, while the effective support spans were both 910 mm. The side length of the core and the lattice web spacing varied from 75 mm, 125 mm, and 175 mm. The geometric details of SLCP specimens are described in Table 1, and the material properties of SLCPs are listed in Table 2. The experimental scheme is shown in Figure 4, where LVDT for the out-of-transverse deflection was set up under the bottom of the SLCP, as shown in Figure 5.

#### 3.1.2. Results and Discussion

The out-of-transverse displacements and face-plate normal stresses of the flexural experiment of SLCPs are discussed here. Figure 6 shows the test load-deflection curves compared to the top, mid, and bottom surface out-of-plane vertical deflections by FEM. Figure 7 shows the load-stress curves on the mid bottom surface. Table 3 lists the deflections and surface normal stresses corresponding to the ultimate bearing capacity.

In accordance with Table 3, the bottom deflection errors between the test and FEM for two-way simply supported SLCPs were within 52–61%. The bottom surface normal stress errors were within 8–22%. However, Figure 6 presents that the test load-deflection curves were evidently elastoplastic. During the elastic stage, the FEM load-deflection curves including the top, mid surface, and bottom were very close to the tests’. Figure 7 presents that the test load-stress curves were weak nonlinear until the ultimate stage. During the whole loading history, the FEM bottom stress curves became close to the tests’.

### 3.2. Case 2: Single-Way Simply Supported SLCPs under Uniformly Concentrated Load

#### 3.2.1. Specimen and Experiment Introduction

Two unidirectional-web and six bidirectional-web reinforced SLCPs, manufactured by vacuum-assisted resin infusion process as in Case 1, were tested under four-point load action. The face sheet was formed by 0°/90°GFRP clothes, and the web sheet was composed of −45°/45°GFRP clothes. That length of all SLCP specimens were 1000 mm, while the effective support spans were both 800 mm, and the width was 300 mm. The side length of the core and the lattice web spacing were 75 mm. The geometric details of SLCP specimens are described in Table 4, and the material properties of SLCPs are listed in Table 5. The experimental scheme is shown in Figure 8, where three LVDTs for the out-of-transverse displacements were set up under the bottom of the SLCP and the top specimen surface at support; the longitudinal and in-plane transverse normal stress on the top and bottom surface center at the midspan.

#### 3.2.2. Results and Discussion

To verify the results with the aforementioned IC_0_FEM-HOZT, the corresponding test results were compared. The out-of-transverse displacements and bilateral normal stresses of the top and bottom midspan face sheets by the flexural experiment for the single-way simply supported SLCPs are discussed here. Figure 9 shows the load-midspan out-of-transverse deflection curves. Figure 10 shows the load-bilateral stress curves on the midspan top and bottom surfaces. Table 6 lists the deflections corresponding to the ultimate bearing capacity. Table 6 lists the deflection corresponding to the ultimate bearing capacity of the test. Table 7 lists that the bilateral midspan normal stresses on the top and bottom surfaces under the ultimate bearing capacity.

In accordance with Table 6, the bottom deflection errors between the test and FEM for single-way simply supported SLCPs were within 1–31%. Based on Table 7, the longitudinal normal stress errors on the top surface were within 11–35%; the transverse errors were within 2–35% except for SXG-2-2-75, which cannot be estimated by the current error method; the longitudinal normal stress errors at the bottom surface were within 14–50%; the transverse errors were within 3–140%. Both the test load-midspan deflection curves and load-stress curves took on elastic, generally, according to Figure 9 and Figure 10. During the whole loading stage, the FEM results including on the top, at the midsurface, and at the bottom were close to the tests’. However, the sudden change of test data at the ultimate loading end showed the nonlinear effect evidently, which can lead to large errors on the relative deflection and normal stress.

## 4. Conclusions

In this study, nine-nodes-isoparametric-quadratic plate element and three-nodes-isoparametric-quadratic beam element were combined to simulate the sandwich lattice composite panel with the improved C_0_ finite element method with the soft compressible core using high-order zigzag theories. The deflections and normal results of SLCPs under the out-of-plane quasi-static loading with FEM were compared to those by the test.

As a whole, the combination method by the improved C_0_ finite plate element and the beam element degenerated from the improved C_0_ finite plate element, based on high-order zigzag theories, is a suitable method for the analysis of the sandwich lattice composite panel. IC_0_FEM-HOZT can avoid the conventional three-dimension hybrid element model composed by cube element, shell element, and beam, which usually may lead to a complicated building program, and the coupling degrees of freedom among the different types of elements. Though some deviation still exists between the calculation results by IC_0_FEM-HOZT and those by test due to the multiple causes such as nonlinear, IC_0_FEM-HOZT has great potential to become an excellent and highly efficient analysis and design tool for the sandwich lattice composite panel, if appropriate modifications are adopted according to the actual work.

## Figures and Tables

**Figure 1 polymers-13-04200-f001:**
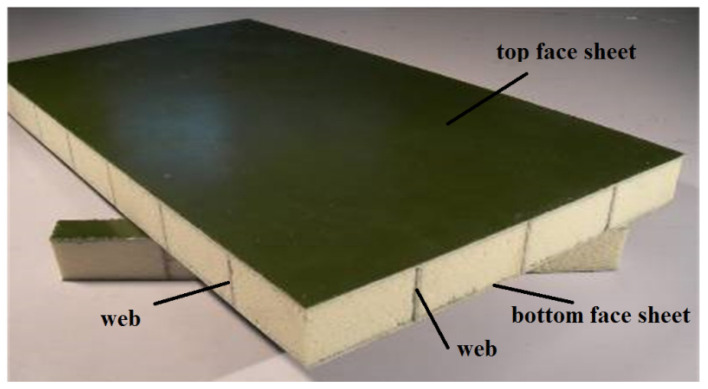
Sandwich lattice composite panel reinforced with crisscross webs.

**Figure 2 polymers-13-04200-f002:**
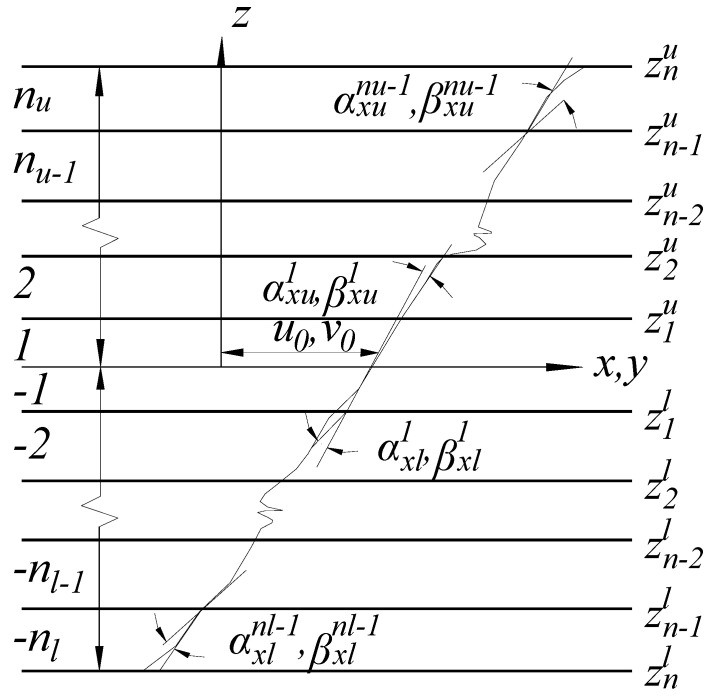
In-plane displacement fields in laminas.

**Figure 3 polymers-13-04200-f003:**
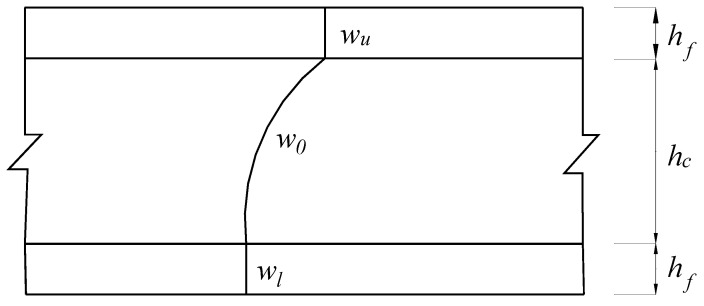
Out-of-transverse displacement fields.

**Figure 4 polymers-13-04200-f004:**
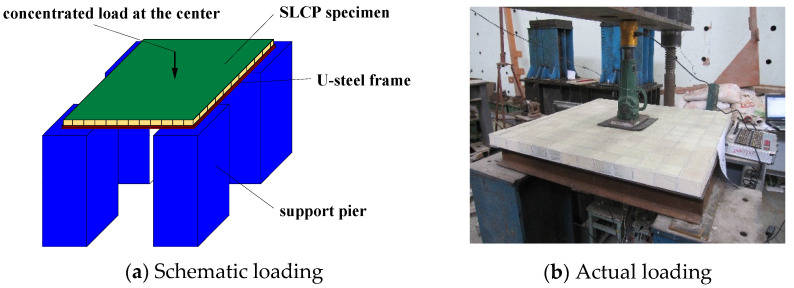
Loading schematic for two-way simply supported SLCPs.

**Figure 5 polymers-13-04200-f005:**
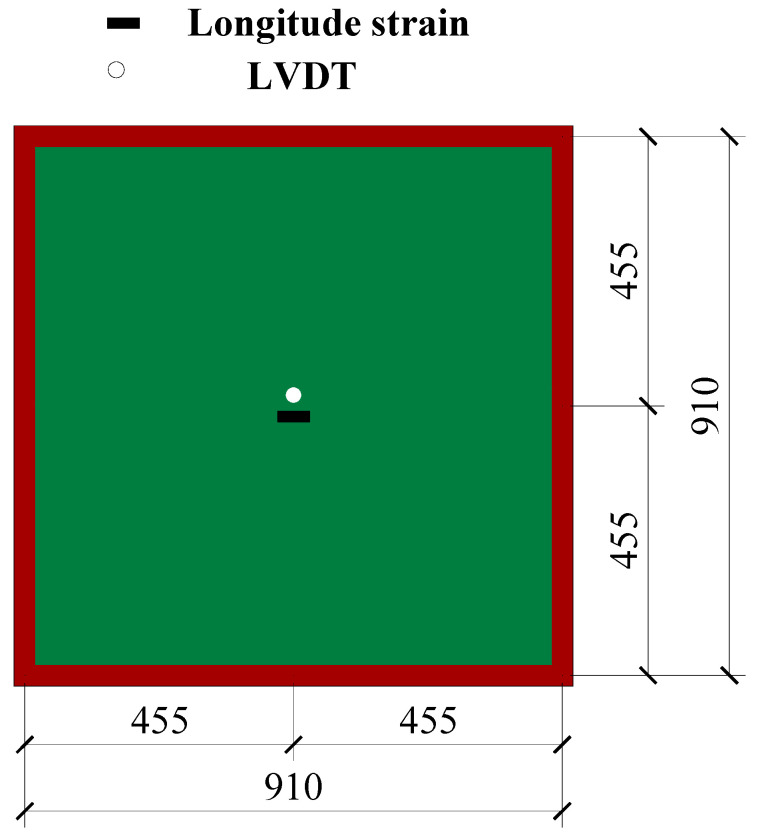
LVDT for the out-of-transverse deflection scheme.

**Figure 6 polymers-13-04200-f006:**
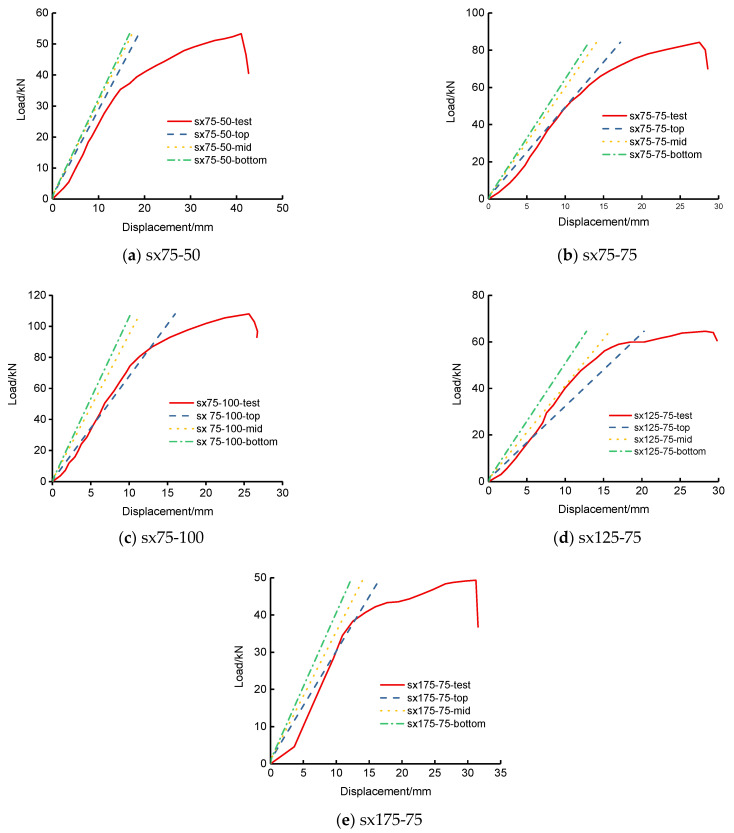
Load-deflection curves under the concentrated load for two-way simply supported SLCPs.

**Figure 7 polymers-13-04200-f007:**
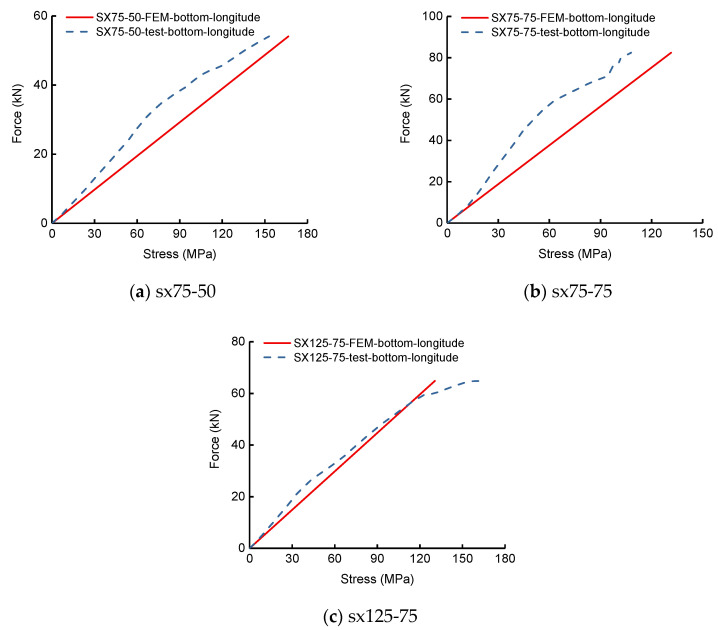
Load-stress curves on the bottom surface of two-way simply supported SLCPs.

**Figure 8 polymers-13-04200-f008:**
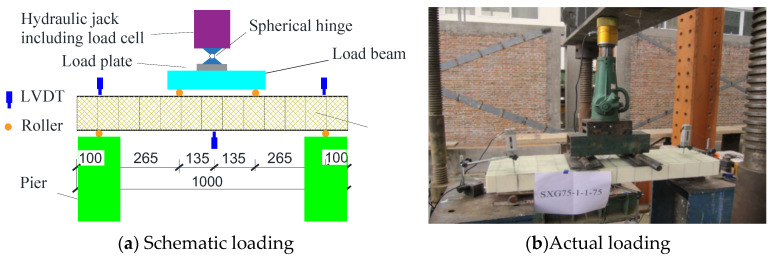
Loading schematic for single-way simply supported SLCPs.

**Figure 9 polymers-13-04200-f009:**
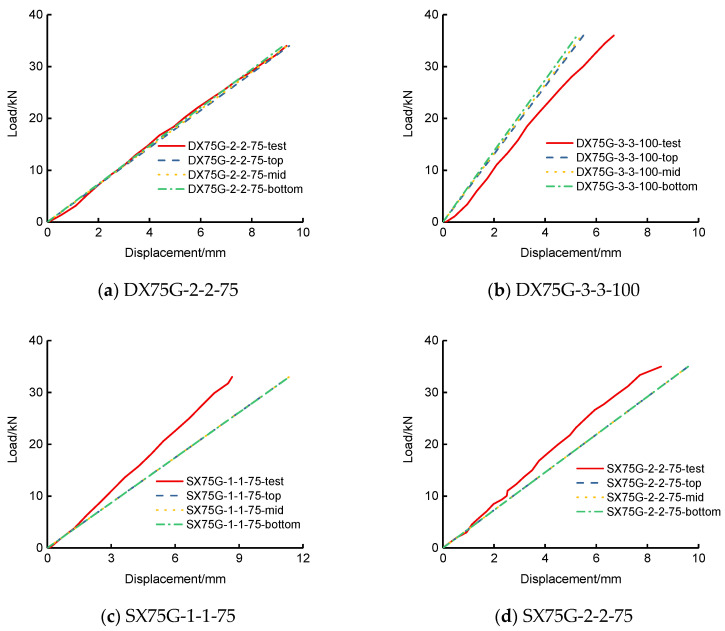
Load-midspan curves under the concentrated load for single-way simply supported SLCPs.

**Figure 10 polymers-13-04200-f010:**
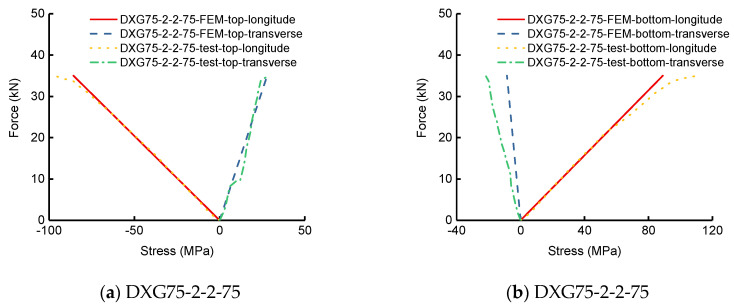
Load-normal stress curves on the mid, top, and bottom surfaces of single-way simply supported SLCPs.

**Table 1 polymers-13-04200-t001:** Details of the specimens.

Specimen	External Dimension/mm			Core Dimension/mm			Face Sheet/mm	Web Sheet/mm	Test Type
	Length	Width	Supported Span	Length	Width	Height	Nominal Thickness	Nominal Thickness	
SX75-50	1000	1000	910	75	75	50	1.60	1.06	L-D, L-S
SX75-75						75			L-D, L-S
SX75-100						100			L-D
SX125-75				125	125	75			L-D, L-S
SX175-75				175	175	75			L-D

Nomenclature: L-D, load displacement; L-S, load stress.

**Table 2 polymers-13-04200-t002:** Material properties.

Items	Components	Values
Elastic modulus,/MPa	Face sheet	20,950
	Web sheet	6410
	Polyurethane foam core	7.17
Shear modulus,/MPa	Face sheet	6714
	Web sheet	5820
	Polyurethane foam core	2.08
Passion ratio	Face sheet	0.15
	Web sheet	0.15
	Polyurethane foam core	0.3

**Table 3 polymers-13-04200-t003:** The mid deflection and normal stress under the ultimate bearing capacity.

Specimen	Ultimate Bearing Capacity/kN	Deflection/mm, δ (Stress/MPa, σ)				Error Ratio between Test and FEM/% |(δ_FEM_ − δ_Test)/_δ_FEM_| × 100% (|(σ_FEM_ − σ_Test)/_σ_FEM_| × 100%)
		Test	FEM			
		Bottom	Top	Mid	Bottom	
SX75-50	53.3	41.0 (153.0)	18.8 (-)	17.3 (-)	16.7 (166.6)	59.2 (8.9)
SX75-75	84.3	27.5 (108.1)	17.2 (-)	14.1 (-)	13.1 (131.5)	52.4 (21.6)
SX75-100	108.1	25.6	16.0	11.4	10.2	60.2
SX125-75	64.6	28.3 (161.7)	20.3 (-)	15.8 (-)	12.8 (130.6)	54.8 (19.2)
SX175-75	49.3	31.2	16.5	14.0	12.2	61.0

**Table 4 polymers-13-04200-t004:** Details of specimens.

Specimen	External Dimension/mm			Core Dimension/mm			Face Sheet/mm	Web Sheet/mm	Web Layer Number		Test Type
	Length	Width	Supported Span	Length	Width	Height	Nominal Thickness	Single Nominal Thickness	Longitudinal	Transverse	
DX75G-2-2-75	1000	300	800	75	75	75	3.20	1.60	2	2	L-D, L-S
DX75G-3-3-100						100			3	3	L-D
SX75G-1-1-75						75			1	1	L-D, L-S
SX75G-2-2-75						75			2	2	L-D, L-S
SX75G-3-3-75						75			3	3	L-D
SX75G-1-1-100						100			1	1	L-D
SX75G-2-2-100						100			2	2	L-D
SX75G-3-3-100						100			3	3	L-D, L-S

Nomenclature: NX, reinforced by single-way webs with the longitudinal direction; SX, reinforced by two-way webs with the longitudinal and transverse direction; G, glass fiber; L-D, load displacement; L-S, load stress.

**Table 5 polymers-13-04200-t005:** Material properties.

Items	Components	Values
Elastic modulus, /MPa	Face sheet	20,950
	Web sheet	8841
	Polyurethane foam core	6.96
Shear modulus, /MPa	Face sheet	6714
	Web sheet	6230
	Polyurethane foam core	2.34
Passion ratio	Face sheet	0.15
	Web sheet	0.15
	Polyurethane foam core	0.3

**Table 6 polymers-13-04200-t006:** The mid deflection under the ultimate bearing capacity.

Specimen	Ultimate Bearing Capacity/kN	Deflection/mm, δ				Error ratio between Test and FEM/% |(δ_FEM_ − δ_Test)/_δ_FEM_| × 100%
		Test	FEM			
		Bottom	Top	Mid	Bottom	
DX75G-2-2-75	34.0	9.4	9.5	9.4	9.2	1.4
DX75G-3-2-100	36.0	6.7	5.5	5.4	5.3	21.4
SX75G-1-1-75	33.0	8.7	11.4	11.4	11.4	31.0
SX75G-2-2-75	35.0	8.5	9.6	9.6	9.6	12.5
SX75G-3-3-75	42.0	9.7	10.4	10.4	10.4	6.8
SX75G-1-1-100	42.0	6.5	8.0	8.0	8.1	24.8
SX75G-2-2-100	44.0	6.3	7.5	7.5	7.5	19.8
SX75G-3-3-100	50.0	8.3	7.5	7.5	7.5	9.7

**Table 7 polymers-13-04200-t007:** The bilateral midspan normal stress under the ultimate bearing capacity.

Specimen	Ultimate Bearing Capacity/kN	Stress/MPa, σ				Error Ratio between Test and FEM/% |(σ_FEM_ − σ_Test)/_σ_FEM_| × 100% (|(σ_FEM_ − σ_Test)/_σ_FEM_| × 100%)	
		Test		FEM			
		Top	Bottom	Top	Bottom	Top	Bottom
DX75G-2-2-75	34.0	−96.6 (28.6)	110.7 (−22.0)	−85.8 (27.9)	89.0 (−8.5)	11.2	19.6
						(2.4)	(138.6)
SX75G-1-1-75	33.0	−90.4 (13.2)	154.0 (−9.2)	−59.1 (13.9)	77.9 (−1.3)	34.6	49.4
						(5.3)	(85.9)
SX75G-2-2-75	35.0	−64.8 (0.0)	77.7 (−8.2)	−82.1 (9.0)	89.0 (−8.5)	26.7	14.5
						(--)	(3.7)
SX75G-3-3-100	50.0	−65.8 (13.9)	2.2 (48.7)	−87.2 (9.1)	91.9 (−7.0)	32.5	40.8
						(34.5)	(114.4)

Annotation: the stress data outside the parenthesis are on behalf of the longitudinal, and those inside denote the transverse.

## Data Availability

Data available in a publicly accessible repository.

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
