# Peer review of "An Improved C0 FE Model for the Sandwich Lattice Composite Panel"

_polymers, 2021, doi:10.3390/polym13234200_

Round 1

Reviewer 1 Report

This paper presented a theoretical model that combines the improved C0 plate element using high order zigzag theories and the beam element degenerated from the plate element for the sandwich lattice panel. The paper can be recommended for publication if these concerns are addressed appropriately:

  1. Why the model could not predict the behaviour after the linear region in Figure 6. Please make some discussions.
  2. In section 3.1.2, the error is quite large please explain clearly
  3. In Fig 10, why the error in the longitudinal direction is higher than that in the transverse direction. Please discuss the result.
  4. Please check the format of the cited reference in the text. For example Line 36-37 on the first page, there is no reference.
  5. The term "etc" in the first paragraph should be removed and please add the conjunction "and" for the last two features "easy design" and multi-forms"
  6. The introduction (line 23-29) can be improved by including the latest achievements on the sandwich panels such as https://doi.org/10.1016/j.compositesb.2021.109301 https://doi.org/10.1016/j.compositesb.2019.107496 https://doi.org/10.1007/s10853-018-3163-x 

Author Response

  1. Why the model could not predict the behaviour after the linear region in Figure 6. Please make some discussions.

Response 1: The improved C0 FE model was proposed to predict the behaviour of sandwich lattice composite panel in elastic statement, as the rigidity matrix was obtained by elastic modulus E (page 6, line 2) in which only linear region was in consideration. According to Fig. 6, the results in linear region were consistent with the test data and the nonlinear behavior of the composite panel would be investigated in future study.

  1. In section 3.1.2, the error is quite large please explain clearly.

Response 2: In experiment, the stiffness degrades gradually due to the development of cracks, resulting in the increase of bottom deflection. As elastic modulus in improved C0 FE model was assumed to be stationary, the results in linear region were consistent with the test data but the errors in nonlinear region was 52%~61%. The nonlinear behavior of the composite panel would be investigated in future study to solve the problem above.

  1. In Fig 10, why the error in the longitudinal direction is higher than that in the transverse direction. Please discuss the result.

Response 3: In single-way simply supported SLCPs, the stress generated by load is mainly distributed along the longitudinal direction and the compressive (tension) cracks developed rapidly, which result in more nonlinear behavior of composite panel leading to higher error than that of transverse direction.

  1. Please check the format of the cited reference in the text. For example Line 36-37 on the first page, there is no reference.

Response 4: Thanks for the suggestion, the number of cited references are [12-15].To make description clear, revised have been made. Please see line 13, page 2.

  1. The term "etc" in the first paragraph should be removed and please add the conjunction "and" for the last two features "easy design" and multi-forms"

Response 5: Thanks for the suggestion. Revision has been made according to the reviewer’s suggestion. Please see line 4, page 2.

  1. The introduction (line 23-29) can be improved by including the latest achievements on the sandwich panels such as https://doi.org/10.1016/j.compositesb.2021.109301 https://doi.org/10.1016/j.compositesb.2019.107496 https://doi.org/10.1007/s10853-018-3163-x

Response 6: Thanks for the suggestion. References above have been added to the revised manuscripts. Please see line 7, page 2; and lines 8-14, page 14.

Reviewer 2 Report

The title of the article caught my attention, but I found nothing new in the content. The adopted element is described only with general formulas, no specific derivations are given - what does the matrix B look like for a plate and beam element? I do not understand what the authors have brought new to science with their publication?

Major comment

Authors must very clearly show what's new in their work. The motivation is fairly clear, but the solution is not clear at all. Whether the presented finite element was implemented and in what programming environment. Has an already implemented element been used in the existing or commercial FEM software? If so, than in what software?

Minor suggestions:

1) who finally is the corresponding author?

2) line 12: "Compared with the actual test results including the mid-span deflections and the surface sheet normal stresses, the outstanding of that method is presented through numeric calculation including the mid-span deflections and the surface sheet normal stresses." - there is a repetition of "the mid-span deflections and the surface sheet normal stresses"

3) line 15:"The results show the model..." - it should be rather: "The results show that the model..."

4) line 17: "complicated building program" what it means?

5) line 29: "The detail Eq. (4) for plate is..." - is where??

6) line 30: "The detail Eq. (4) for beam is..." - is where??

7) there are many incomprehensible phrases in the text that result from poor translation - the manuscript has to be carefully revised by native speaker

8) the font used in all equations is too big and looks really awkward

Author Response

Reviewer #2:

The title of the article caught my attention, but I found nothing new in the content. The adopted element is described only with general formulas, no specific derivations are given - what does the matrix B look like for a plate and beam element? I do not understand what the authors have brought new to science with their publication?

Response: Thank you very much for your constructive and valuable comments. We have revised our manuscript according to the comments and have marked the changes in blue in the revised version. Please see the details in the revised manuscript. The responses to the reviewer’s comments are as follows:

Major comment

Authors must very clearly show what's new in their work. The motivation is fairly clear, but the solution is not clear at all. Whether the presented finite element was implemented and in what programming environment. Has an already implemented element been used in the existing or commercial FEM software? If so, than in what software?

Response: Thanks for this suggestion. The new in this work is that using high order zigzag theories, combing nine-nodes-isoparametric-quadratic plate element and three-nodes-isoparametric-quadratic beam element to simulate the sandwich lattice composite panel with the improved C0 finite element method with the soft compressible core, please see lines 13-18, page 4. The proposed finite element method is still in theoretical calculation and will be realized in future studies.

Minor suggestions:

  1. Who finally is the corresponding author?

Response 1: There are two corresponding author, Junqing Hong and Hai Fang.

  1. Line 12: "Compared with the actual test results including the mid-span deflections and the surface sheet normal stresses, the outstanding of that method is presented through numeric calculation including the mid-span deflections and the surface sheet normal stresses." - there is a repetition of "the mid-span deflections and the surface sheet normal stresses"

Response 2: Thanks for this suggestion, modifications have been made and the repetition has been deleted. Please see lines 11 and 12, page 1.

  1. Line 15:"The results show the model..." - it should be rather: "The results show that the model..."

Response 3: Thanks for this suggestion. The "The results show the model" has been replaced by "The results show that the model”, please see line 12-13, page 1.

  1. Line 17: "complicated building program" what it means?

Response 4: The “complicated building program” means complex processes in the process of finite element modeling and calculation. To make description clear, the sentence above has been replaced by “complex program establishment”. Please see line 15. Page 1; and line 11, page 13.

  1. Line 29: "The detail Eq. (4) for plate is..." - is where??

Response 5: Thanks for this suggestion, Eq. (4) has been supplied, please see line 4, page 6.

  1. Line 30: "The detail Eq. (4) for beam is..." - is where??

Response 6: Response: Thanks for this suggestion, Eq. (4) has been supplied, please see line 6, page 6.

  1. There are many incomprehensible phrases in the text that result from poor translation - the manuscript has to be carefully revised by native speaker

Response 7: Thanks for this suggestion. We have found an English language editor from American Journal Experts to check our manuscript to avoid any further grammar, syntax or spelling mistakes. We believe that the English writing is now at an acceptable level.

  1. The font used in all equations is too big and looks really awkward

Response 8: Thanks for this suggestion, and the font used in all equations are resized to size 12, same as normal the font size in the text.

Reviewer 3 Report

An improved C0 FE model for the sandwich lattice composite panel – J. Hong et al.

General Comments: An improved FE model for sandwich composite is proposed wherein, the authors have attempted to incorporate high order zigzag theory and degenerated beam element. The manuscript editing for English style and grammar before it can be recommended for publication.  My specific comments can be found below.

Specific Comments:

  • Line 13: Outstanding feature...
  • Was the core homogenized in the model? How did the authors take into account the webs?
  • The bottom deflection errors were noted well above 52%, authors must clarify this discrepancy observed between test and model.
  • Is this model only valid in the elastic regime?
  • Table 3 indicates high error values in stress, please clarify.
  • Does the model also take into account debonding? It is quite unclear how this model will predict ultimate load without predicting face/core disbond initiation.

Author Response

Response to Reviewer 3 Comments

General Comments: An improved FE model for sandwich composite is proposed wherein, the authors have attempted to incorporate high order zigzag theory and degenerated beam element. The manuscript editing for English style and grammar before it can be recommended for publication.  My specific comments can be found below.

Response: Thanks for this suggestion. We have found an English language editor from American Journal Experts to check our manuscript to avoid any further grammar, syntax or spelling mistakes. We believe that the English writing is now at an acceptable level.

Specific Comments:

  1. Line 13: Outstanding feature...

Response 1: Thanks for this suggestion. Revision has been made according to reviewer’s suggestion. Please see line 12, page 1.

  1. Was the core homogenized in the model? How did the authors take into account the webs?

Response 2: The core was homogenized in the model, please see lines 2-4, page 6. The webs was regarded as the part of the core material, as the spatial occupied core areas by the web volumes can be substracted by the same virtualized volume web with the core material, or even omitted since the web is very thin and the elastic modulus is far high than that of the core material, please see lines 20-22, page 4.

  1. The bottom deflection errors were noted well above 52%, authors must clarify this discrepancy observed between test and model.

Response 3: In experiment, the stiffness degrades gradually due to the development of cracks, resulting in the increase of bottom deflection. As elastic modulus in improved C0 FE model was assumed to be stationary, the results in linear region were consistent with the test data but the errors in nonlinear region was above 52%. The nonlinear behavior of the composite panel would be investigated in future study to solve the problem above.

  1. Is this model only valid in the elastic regime?

Response 4: The improved C0 FE model was proposed to predict the behaviour of sandwich lattice composite panel in elastic statement, as the rigidity matrix was obtained by elastic modulus E (line 2, page 6,) in which only linear region was in consideration. The errors in nonlinear region was above 52% between the test and model, therefore, the nonlinear behavior of the composite panel would be investigated in future study.

  1. Table 3 indicates high error values in stress, please clarify.

Response 5: As the compressive (tension) cracks developed rapidly after the linear region resulting in more nonlinear behavior of composite panel, which leading to higher error values in stress.

  1. Does the model also take into account debonding? It is quite unclear how this model will predict ultimate load without predicting face/core disbond initiation.

Response 6: According to references [12-15], debonding did not happened before the ultimate load due to the limitation of lattice-webs. Therefore, disbond initiation was not considered in this paper, and the ultimate load was obtained by high order zigzag theories and the beam element degenerated from the plate element.

Round 2

Reviewer 1 Report

The manuscript has been significantly improved after the revision. The reviewer recommends it for publication. 

Reviewer 2 Report

As the Authors addressed all my critical remarks, I suggest to accept the article as it is.

Reviewer 3 Report

The current version is fine; although the explanation on debond initiation is not satisfactory. I would suggest highlighting in the text that the proposed model works in the elastic regime and disbond initiation is not predicted. Debonding is an interface phenomenon and has nothing to do with the lattice cores. Yes, they might reinforce it but, invariably debond is directly attributed to face/core interface fracture toughness. There are ample references in that regard in the literature doi.org/10.1177/1099636218788223, doi.org/10.1177/1099636218777964, doi.org/10.1177/1099636220909820 which describes the characterization of disbond initiation in typical sandwich structures.